# A cortico-collicular pathway for motor planning in a memory-dependent perceptual decision task

Chunyu A. Duan [1,5 ✉], Yuxin Pan [1,2,5], Guofen Ma[3], Taotao Zhou[1], Siyu Zhang [3] & Ning-long Xu [1,2,4 ✉]

Survival in a dynamic environment requires animals to plan future actions based on past sensory evidence, known as motor planning. However, the neuronal circuits underlying this crucial brain function remain elusive. Here, we employ projection-specific imaging and perturbation methods to investigate the direct pathway linking two key nodes in the motor planning network, the secondary motor cortex (M2) and the midbrain superior colliculus (SC), in mice performing a memory-dependent perceptual decision task. We find dynamic coding of choice information in SC-projecting M2 neurons during motor planning and execution, and disruption of this information by inhibiting M2 terminals in SC selectively impaired decision maintenance. Furthermore, we show that while both excitatory and inhibitory SC neurons receive synaptic inputs from M2, these SC subpopulations display differential temporal patterns in choice coding during behavior. Our results reveal the dynamic recruitment of the premotor-collicular pathway as a circuit mechanism for motor planning.

[1] Institute of Neuroscience, State Key Laboratory of Neuroscience, CAS Center for Excellence in Brain Science and Intelligence Technology, Chinese Academy of Sciences, Shanghai, China. [2] University of Chinese Academy of Sciences, Beijing, China. [3] Collaborative Innovation Center for Brain Science, Department of Anatomy and Physiology, Shanghai Jiao Tong University School of Medicine, Shanghai, China. [4] Shanghai Center for Brain Science and Brain-Inspired Intelligence Technology, Shanghai, China. [5] These authors contributed equally: Chunyu A. Duan, Yuxin Pan. ✉email: cduan@ion.ac.cn; xunl@ion.ac.cn

To survive in a dynamic environment with volatile sensory cues, it is often crucial for animals to maintain choice-related information in the gap between sensation and action. How the brain bridges past events with future actions is a fundamental question in neuroscience and has been studied in animals using delayed-response motor planning tasks[1,2]. In these tasks, animals plan a response based on transiently presented sensory cues, but then need to withhold action and maintain this motor plan during a delay period without sensory cues. Using this classic paradigm, neural activity in the premotor cortex is shown to be critical for planning delayed responses in macaques[3,4], rats[2,5,6], and mice[7–10].

Premotor cortex forms complex networks with many cortical and subcortical areas. Recent work in head-fixed mice has started to reveal distinct contributions of premotor subpopulations based on their projection targets, such as the thalamus[11], brainstem[7,12], and cerebellum[13]. Thalamus-projecting premotor neurons contribute to the maintenance of persistent delay activity[11,12]; whereas brainstem-projecting neurons are more involved in motor execution[12]. Another major subcortical target of the premotor cortex is the midbrain superior colliculus (SC). Non-overlapping thalamus-projecting and brainstem-projecting premotor neurons both collateralize and project to the SC deep layers[12]. The SC also receives inputs from many other sensory and motor regions and sends ascending and descending projections to various subcortical areas[14–16], serving as a potential hub region that links sensation and action. Indeed, the SC has been implicated in planning and executing orienting responses in macaques[17–20] and rats[5,21–24].

Even though both the premotor cortex and SC have been shown to be involved in motor planning, the vastly complex networks associated with the premotor cortex and SC obscures the underlying circuit logic. For instance, it remains unclear whether the premotor cortex and SC contribute to motor planning via separate channels in parallel, or via a direct projection pathway; and whether SC simply inherits the motor commands from the premotor cortex or actively participates in motor planning during the time gap between sensory decisions and motor execution. Furthermore, it is also unclear how specific subtypes of neurons that form the mesoscale connectivity between the two distant regions contribute to the information routing during motor planning.

To investigate the information routing in the direct premotor-collicular pathway and its precise contribution to motor planning, here we developed a delayed-response task in head-fixed mice parametrizing the memory demands by varying the stimulus difficulty and delay duration and employed projection-specific methods to examine the projections from the secondary motor cortex (M2) to SC during memory-guided directional licking. Using multisite optogenetic perturbations, we found concurrent involvement of M2 and SC in planning licking responses. We then conducted in vivo two-photon calcium imaging from SC-projecting M2 neurons and found that M2 sends progressively stronger choice-related information to SC, which correlates with animals' response time. Furthermore, we used chemogenetic manipulation to specifically disrupt the M2-SC information flow by inhibiting M2 axonal terminals in SC and found preferential impairment of mice's performance in more demanding conditions with difficult stimuli and long-delay durations, supporting a critical role of the M2-SC pathway in decision maintenance. Finally, we used circuit mapping and cell-type-specific photometric recordings to show that the choice-related information transmitted from M2 underwent differential further processing by the excitatory and inhibitory neurons in SC, with the excitatory neurons more faithfully maintaining the contralateral choice information over time and the inhibitory neurons displaying

more dynamic and heterogeneous choice coding. Our results reveal that the M2-SC pathway plays a critical role in maintaining decision information over time, representing a mesoscale connectivity-based circuit mechanism underlying a fundamental cognitive function.

## Results

**M2-dependent memory-guided perceptual decision task**. We trained head-fixed mice to perform a delayed-response auditory discrimination task, with varying perceptual difficulty and delay duration (Fig. 1). On each trial, mice were presented with auditory click trains of different rates (20–125 Hz). Click rates lower than 50 clicks/s indicate water reward from the left lick port; while higher lick rates indicate reward from the right port (Fig. 1a). Stimulus–response mapping was counterbalanced across animals. Mice need to discriminate the sensory cues, form and maintain a decision during a variable delay period, and respond by licking one of the two lick ports after an auditory "go" cue (Fig. 1b, c). Trials where animals lick before the "go" cue are considered violation trials and excluded from further analyses. Most previous studies using delayed-response tasks in head-fixed mice presented only two sensory stimuli to elicit two motor responses[8,10,25,26], unable to vary stimulus difficulty. Here, we varied both perceptual difficulty (Fig. 1d) and delay duration (Fig. 1e, f) on a trial-by-trial basis, which allows us to examine the circuit mechanisms for motor planning with varying degrees of memory demand. As delay duration increases, mice's response time (RT) decreased from $567.1 \pm 25.4$ ms to $448.1 \pm 26.8$ ms (mean ± s.e.m. across animals, bootstrapped $P < 0.01$ across 8 mice; Fig. 1f), consistent with a motor preparation process[27]. We did not find a consistent effect of stimulus difficulty on RT ($P > 0.05$ across eight mice for any delay duration).

Converging evidence in head-fixed mice has implicated M2 as a cortical substrate for planning directional licking responses[7,8,10]. We thus tested whether our delayed-response auditory discrimination task also requires M2 activity using chemogenetic silencing (Supplementary Fig. 1). We expressed hM4Di, a designer receptor exclusively activated by designer drug (DREADD)[28], in bilateral M2 neurons using a viral vector (AAV-Syn-hM4D(Gi)-mCherry), and conducted intraperitoneal (IP) injection of either Clozapine-N-oxide (CNO) or saline before each behavioral session (Supplementary Fig. 1a, b). The effect of inactivation was quantified by comparing task performance in CNO sessions to that in the corresponding saline control sessions ("Methods"). M2 inactivation impaired animals' choice accuracy ($P < 10^{-3}$, bootstrap, $n = 15$ sessions; Supplementary Fig. 1c, d), increased animals' RT on correct trials (RT increase $= 128.1 \pm 33.8$ ms, $P < 10^{-3}$; Supplementary Fig. 1e, f), and increased animals' miss rate ($P < 10^{-3}$; Supplementary Fig. 1g), with no difference between easy versus hard trials ($P$'s $> 0.05$). These results were not due to any nonspecific effects of CNO IP injection alone (Supplementary Fig. 1i–l), and suggest that our task is M2-dependent.

**Involvement of SC in memory-guided directional licking.** Although the midbrain SC has been extensively studied in the context of orienting behavior[14,15,29,30], studies of SC's functions for choice effectors beyond orienting are relatively rare[31,32], and the precise contribution of the direct M2-SC pathway in motor planning remains unclear. To identify the anatomical M2-SC pathway potentially involved in memory-guided directional licking, we conducted anterograde tracing from M2 projection neurons to label the subregion of SC directly downstream of M2. We found dense axon terminals from M2 in the anterior lateral part of SC (Fig. 2a), a subregion of SC previously implicated in

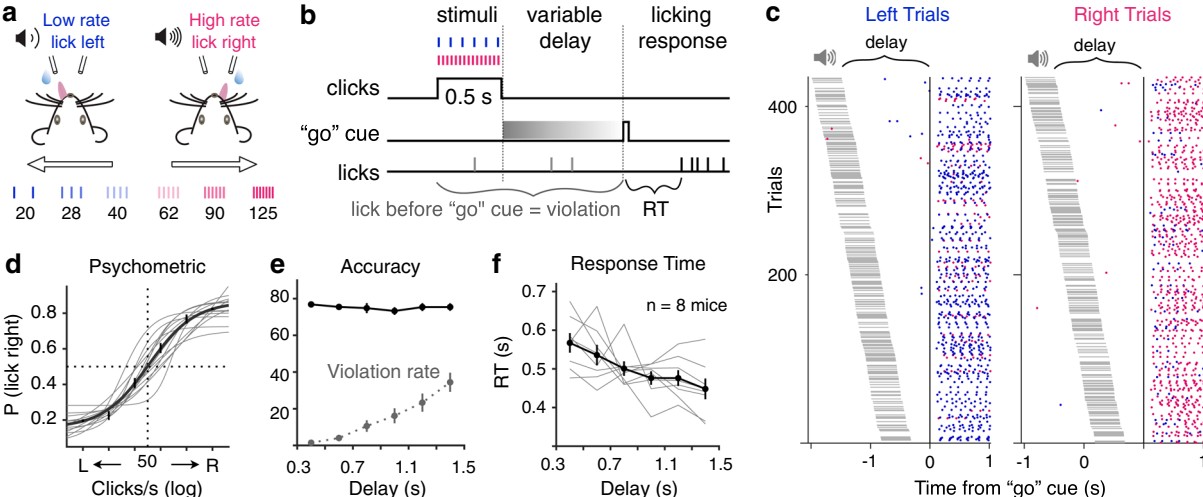

**Fig. 1 A memory-guided click rate discrimination task in head-fixed mice. a** Schematic of behavioral paradigm. **b** Temporal structure of a task trial. A variable delay period sampled from a uniform distribution (0.3–1.5 s) is randomly assigned for each trial. Trials with premature licking are marked as violations and excluded from further analyses. For non-violation trials, the response time (RT) is defined as the time between the onset of the "go" cue and animals' first lick. **c** Lick raster during an example session. Left licks (blue dots) and right licks (magenta dots) are plotted for left and right trials, aligned to the "go" cue, and sorted by delay durations. Timings of the sound are marked by gray horizontal lines. **d** Sigmoid fit of psychometric performance. Thick line, mean fit; thin lines, individual mice ($n = 14$); error bars, s.e.m. across all trials. **e** Accuracy (solid black) and violation rate (dotted gray) as a function of delay duration. Mean and s.e.m. across mice ($n = 8$ mice). **f** Response time (RT) decreases as a function of delay duration. $P = 0.004$ for shortest vs longest delay duration, two-sided bootstrap test, $n = 8$ mice. Black thick lines, mean and s.e.m. across mice; gray thin lines, normalized median RTs for individual mice.

voluntary drinking[32]. Past work showed that M2 neurons that directly project to the brainstem reticular nuclei are involved in controlling directional licking[7]. Here, we labeled SC somas that are downstream of M2 using anterograde transsynaptic viral tracing[33] and found that these SC neurons also project to brainstem reticular nuclei (Fig. 2b and Supplementary Fig. 2), specifically the gigantocellular reticular nucleus (Gi), where neurons[34] and their projections[35,36] have been implicated in rhythmic licking and tongue movement. These anatomical tracing experiments identify the lateral SC as a potential key node in the cortico-subcortical pathway controlling licking responses.

To test how M2 and SC are dynamically recruited for planning and executing decision-related licking responses, we separately inactivated M2 or SC neurons using optogenetics during different trial epochs, including the intertrial interval (ITI), stimulus, delay, or response period (500 ms inactivation for all epochs; Fig. 2c–e and Supplementary Fig. 3). In each behavioral session, photostimulation was delivered unilaterally on 30% of randomly chosen trials to inhibit local activity (Fig. 2c, d; "Methods"). Performance on inactivation trials was compared to that on control trials in the same session. Different optogenetic conditions (ITI, stimulus, delay, or response period) were interleaved for all sessions to control for behavioral fluctuations across days. We used a generalized linear mixed model (GLMM) to quantify the effect of optogenetic inactivation on animals' behavior while taking into account between-subject variance ("Methods"). We found that unilateral M2 inactivation led to a significant ipsilateral bias (contralateral impairment) when the laser was delivered during the stimulus (ipsi bias = $9.9 \pm 2.5\%$, $P < 10^{-3}$, GLMM), delay ($14.4 \pm 2.4\%$, $P < 10^{-8}$), or response period ($5.8 \pm 2.7\%$, $P < 0.05$), but not during the ITI ($-1.1 \pm 3.3\%$, $P = 0.62$; Fig. 2f–h). The greatest impairment occurred when M2 was inactivated during the delay period (corrected single-step multiple comparison test of GLMM; Fig. 2h), suggesting that M2 plays a more important role during motor planning than during motor execution.

Because SC is classically associated with motor functions[29], it is thus conceivable that SC may be more involved in motor

execution than motor planning. On the contrary, similar to the M2 inactivation results, we found the greatest impairment of inhibiting SC activity following delay period inactivation (ipsi bias = $23.3 \pm 2.1\%$, $P < 10^{-15}$, GLMM), with significant effects also found after the stimulus ($16.0 \pm 2.3\%$, $P < 10^{-3}$) or response period inactivation ($16.9 \pm 2.3\%$, $P < 10^{-5}$), but not during the ITI ($4.2 \pm 3.6\%$, $P = 0.36$; Fig. 2i–k). Slopes of the psychometric functions were not significantly changed after unilateral M2 or SC inactivations ($P$'s > 0.05, GLMM). These results suggest that inducing a hemispheric imbalance of M2 or SC activity, especially during the delay period, is sufficient to affect animals' performance in the memory-guided licking behavior.

To distinguish whether SC plays a permissive role relaying motor command from M2 to brainstem or a more active role, further processing cortical information to facilitate motor planning, we used a within-subject multisite optogenetic perturbation approach to compare M2 inactivation with simultaneous inactivation of both M2 and SC during different behavioral epochs (Fig. 2l–p and Supplementary Fig. 4). Recordings using optrode in awake mice showed that the optimal laser stimulation parameters are different for M2 and SC neurons (Supplementary Fig. 3d–g; "Methods"). To ensure the temporal resolution of optogenetic silencing, we used the lower stimulation parameters for the simultaneous inactivation experiment, which resulted in complete inhibition of M2 activity and partial inhibition of SC activity. Therefore, the simultaneous inactivation can only be fairly compared to M2 inhibition alone.

We used a single GLMM to fit the optogenetic perturbation effects across three types of brain region inactivation and four task epochs (ITI inactivation as control), with click rates, region, and epoch as fixed effects and random effects across animals (Supplementary Fig. 4a, c, Model 1, red curves; "Methods"). Single-step multiple comparisons of GLMM results revealed that simultaneous inactivation of M2 and SC resulted in a significantly greater bias than M2 inhibition alone ($P < 10^{-3}$; Fig. 2o and Supplementary Fig. 4b, c), suggesting that SC makes additional contributions to planning directional licking responses beyond

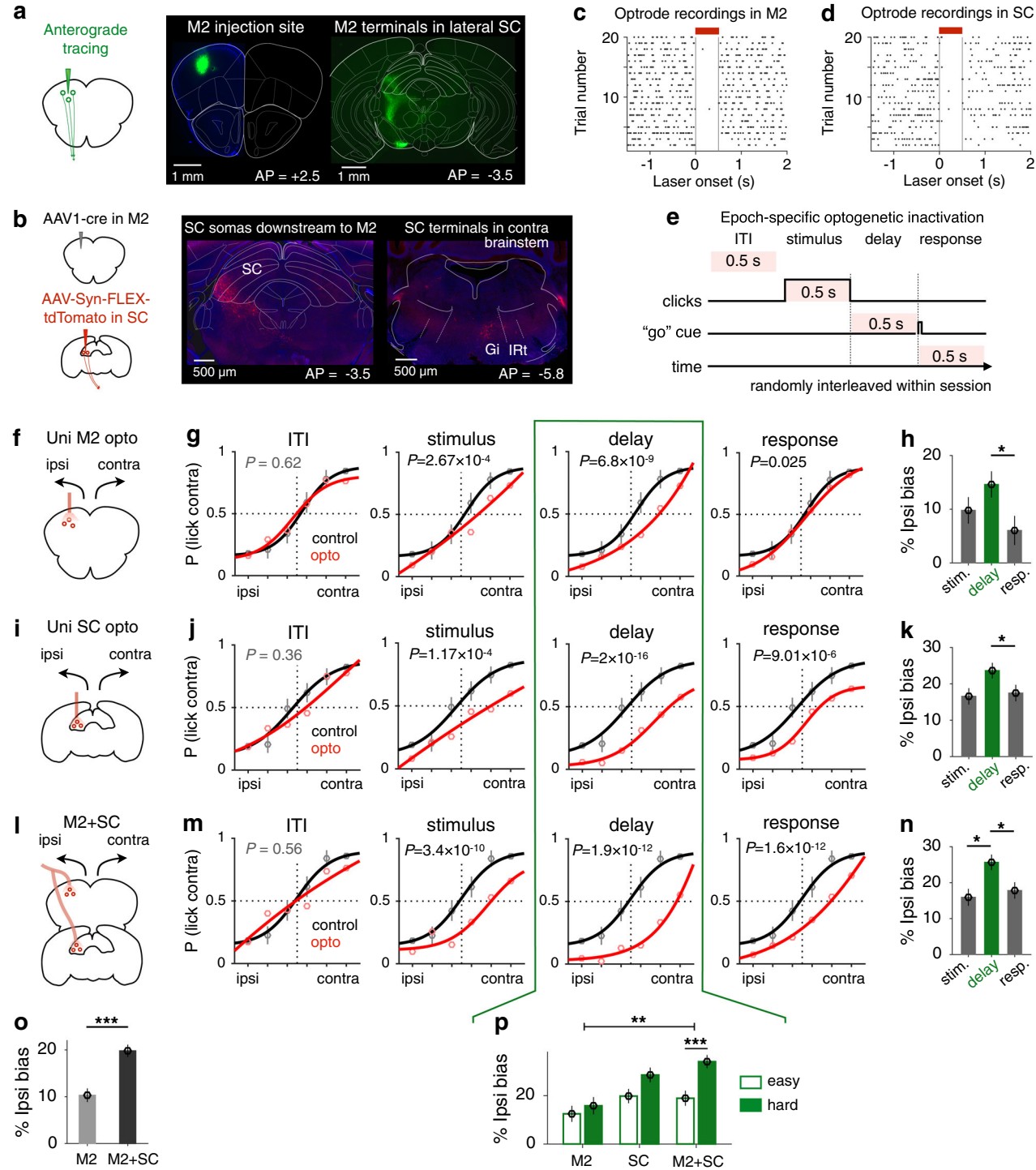

relaying commands from M2. Moreover, this analysis confirmed that M2 and SC are preferentially involved during the delay period, compared to the stimulus ($P < 0.01$) or the response period ($P < 10^{-3}$, GLMM, Supplementary Fig. 4c). Finally, we tested whether optogenetic perturbation of the premotor-collicular circuit affected difficult trials more compared to easy trials by adding a three-way interaction term (region × epoch × difficulty) to the GLMM (Supplementary Fig. 4a, c, Model 2, blue curves). Including this interaction term significantly improved the overall model fit ($P < 0.05$, likelihood ratio test, Chisq). The difficulty effect was significant only when M2 and SC neurons were both silenced during the delay period ($P < 10^{-3}$; Fig. 2p and Supplementary Fig. 4c), but not in any other

condition ($P$'s $> 0.05$, overlapping red and blue curves in Supplementary Fig. 4c), consistent with the possibility that M2 and SC make concurrent contributions to decision maintenance.

**Projection-based imaging reveals dynamic coding of choice-related information in the M2-SC pathway**. To test if the M2-SC pathway carries task-relevant information, we performed projection-specific in vivo two-photon calcium imaging from a subset of M2 neurons that project to the SC with retrograde expression of GCaMP6s (Fig. 3a; "Methods")[37]. We imaged populations of apical dendritic trunks (552) and somas (75) of SC-projecting M2 neurons in randomly selected fields of views

**Fig. 2 M2 and SC's involvement in memory-guided directional licking. a** Anterograde tracing from M2 reveals projection terminals in lateral SC. **b** SC somas downstream of M2 are labeled using transsynaptic virus (left). These SC neurons project to the contralateral brainstem (right). Gi gigantocellular reticular nucleus, IRt intermediate reticular formation. **c** Acute extracellular recordings in awake mice to confirm optogenetic inactivation effect in M2. Spike activities in M2 neurons are aligned to laser onset over trials. The laser illumination period (0.5 s) is marked by the red bar. **d** Similar to **c**, for SC. **e** In each session, optogenetic inactivation during different behavioral epochs were randomly interleaved with no-laser trials (70%). The duration of laser stimulation was kept constant for all epochs (500 ms). **f** Optogenetic inactivation of unilateral M2 (left or right). **g** Data (circles, means and s.e.m. across 5219 control and inactivation trials concatenated across 12 sessions) and 4-parameter sigmoid fit (lines, for visualization only) for M2 inactivations during the intertrial interval (ITI), stimulus, delay, or response period, compared to control trials. $P$ values report ipsilateral bias after optogenetic inactivation, based on two-sided $z$ test on the coefficients estimated using the generalized linear mixed model fit (GLMM, logistic fit), lme4 package in R. No adjustments were made for multiple comparisons. **h** Mean ipsilateral bias caused by M2 inactivation during different behavioral epochs; error bars, s.e.m. across trials. $n = 397, 385, 339$ inactivation trials for stimulus, delay, and response epochs. Between-epoch $P$ values report corrected single-step multiple comparison Tukey's test of GLMM results. $*P = 0.029$. **i–k** Optogenetic inactivation of unilateral SC (5747 control and inactivation trials concatenated across 12 sessions), similar to **f–h**. $n = 452, 452, 444$ inactivation trials for stimulus, delay, and response epochs. $*P = 0.047$ in **k**. **l–n** Simultaneous optogenetic inactivation of M2 and SC on the same side (5425 control and inactivation trials concatenated across 12 sessions), similar to **f–h**. $n = 408, 414, 415$ inactivation trials for stimulus, delay, and response epochs. $*P = 0.037$ for stimulus versus delay period, $*P = 0.05$ for delay versus response period. **o** Mean ipsilateral bias after M2 inactivation or simultaneous inactivation of M2 and SC; error bars, s.e.m. across 1121 inactivation trials for M2 and 1237 inactivation trials for M2 and SC, across all epochs. Between-region $P$ values report corrected single-step multiple comparison Tukey's test of GLMM results. $***P = 10^{-3}$. **p** A difficulty-dependent impairment was observed only when M2 and SC neurons were both inactivated during the delay period (GLMM). Error bars, s.e.m. across trials, $n = 217$ easy trials and 168 hard trials for M2 inactivation; $n = 253$ easy trials and 199 hard trials for SC inactivation; $n = 235$ easy trials and 179 hard trials for M2 and SC inactivation. $**P = 0.008$; $***P = 10^{-4}$, two-sided bootstrap tests.

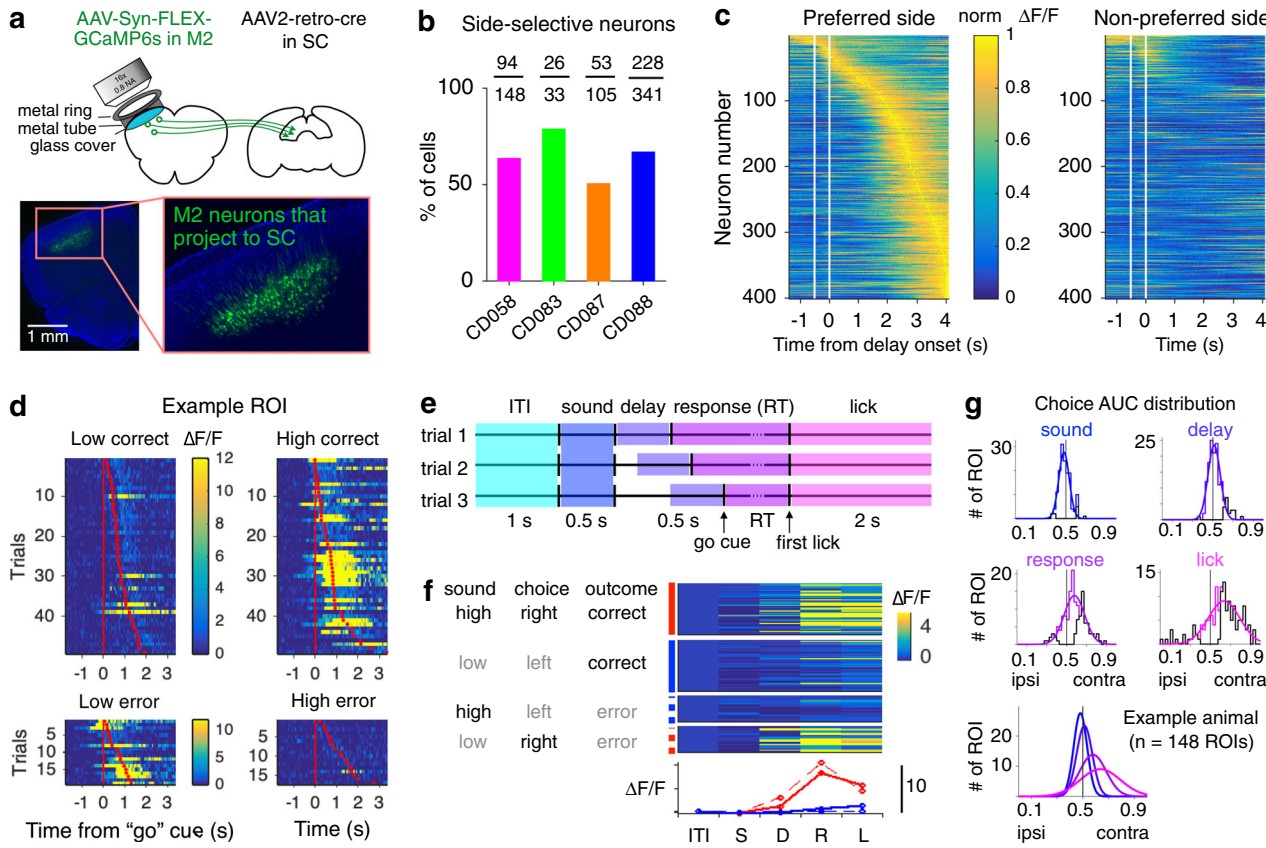

**Fig. 3 Projection-based imaging reveals choice-related information in the M2-SC pathway. a** Design for projection-based imaging experiments (top) and example histology that shows the selective expression of GCaMP6s in SC-projecting M2 neurons (bottom). **b** Percentage and the actual number of side-selective neurons for different animals. **c** Dynamics of all side-selective neurons separated by the choice of the animals, sorted by the time of peak activity on the preferred side. White lines mark the timing of the sound period. **d** Responses (ΔF/F) of an example choice-selective ROI from one imaging session. Each row shows the response for one trial. Red ticks mark RT across trials. Each of the four panels shows one trial type. Diagonal panels correspond to the same type of choice. **e** Schematic that shows the relevant time window of responses extracted for different behavioral epochs. **f** Simplified response profile for the same ROI as **d**. Top, average activity across frames in each behavioral epoch for individual trials. Each block of trials corresponds to one of the four trial types. Bottom, red and blue traces show the average ΔF/F responses for right and left trials (solid for correct, dashed for error trials) across behavioral epochs. **g** Histograms and Gaussian fits (smooth lines) to distributions of AUC values of all the ROIs from one example animal ($n = 148$ ROIs), during the sound, delay, response, and lick epochs. Black lines show individually significant neurons. Right, distributions across epochs are plotted together to reveal a gradual increase of contralateral selectivity over time.

(627 ROIs, 9 sessions, 4 mice) during task performance. We combined the datasets from dendritic trunks and somas because previous studies have reported global calcium signals highly correlated across apical dendritic trunks and somas in layer 5 cortical neurons under awake conditions[38–41], and we found similar activity and selectivity patterns between these compartments (Supplementary Fig. 6a–f). Among all imaged SC-projecting neurons, we found 64.0% (401/627) to be side-selective ($P < 0.01$ for any behavioral epoch; Fig. 3b–c; "Methods"). Individual neurons showed heterogeneous choice-related activity during pre-movement and movement periods (Fig. 3d and Supplementary Fig. 5a). To compare choice selectivity across different behavioral epochs for individual neurons, we conducted receiver operating characteristic (ROC) analysis[42] on responses during the relevant time window for each behavioral epoch (Fig. 3e, f); and calculated the area under the ROC curve (AUC) to indicate discrimination between contra- versus ipsilateral choices ("Methods"). At the population level (including all imaged SC-projecting neurons), we compared distributions of AUC values and observed a progressively stronger contralateral preference as the trial unfolds, from the sound epoch to the delay epoch, and finally the response epoch ($P < 0.01$ for each pair of epochs in the example animal, $P < 10^{-3}$ for each pair of epochs for data across all animals, permutation tests; Fig. 3g and Supplementary Fig. 5b). This gradual increase of contralateral choice selectivity during the trial reflects an emerging motor command in the premotor-collicular pathway. In comparison, generally labeled layer 5 pyramidal neurons in M2 with no projection specificity (736 ROIs, 7 sessions, 2 mice) did not show the gradual evolution of the population choice selectivity toward the contralateral side (Supplementary Fig. 6g–i; "Methods"), suggesting that such progressive dynamics in population selectivity is a cell-type-specific functional property.

The projection-based imaging method enabled us to sample a large number of neurons simultaneously with cell-type specificity ($69.7 \pm 23.4$ ROIs/session), which in turn allowed us to decode animal's choice on a single-trial basis from population activity in the M2-SC pathway. We used cross-validated linear classifiers to decode the amount of choice or sensory information during different behavioral epochs (Fig. 4a, b; "Methods"). We found that population-level delay activity contained significantly above-chance choice information in most sessions (6/9 sessions, $P < 10^{-3}$; Fig. 4c). In addition, the amount of choice information was significantly higher than the amount of sensory information during the delay (bootstrap, $P < 10^{-3}$). Video analyses of mice's tongue movements suggest that the delay-period neural activity was not due to incidental, undetected licks (Supplementary Fig. 7a–e; "Methods"). Consistent with single-neuron analyses, population decoding results also showed progressively stronger choice information as the trial unfolds. A repeated-measure two-way ANOVA on decoding accuracy revealed a significant main effect of choice versus sensory information ($P < 10^{-3}$), a significant main effect of task epoch ($P < 10^{-10}$), and a significant effect of interaction ($P < 10^{-5}$; Fig. 4c). The dynamic evolution of the population choice selectivity in the SC-projecting M2 neurons is consistent with previously reported coding dynamics in the projection-defined premotor cortical neurons, where individual neurons exhibited ramping activity and population-averaged activity showed gradually increased contralateral selectivity over time[7].

To examine the relationship between the amount of accumulated choice information in the M2-SC pathway and an animal's behavior on a trial-by-trial basis (Fig. 4d), we evaluated the correlation between delay-period activity and an animal's response time (RT). We found that shorter RT tended to happen on trials with longer delay durations ($r = -0.74$, $P < 10^{-4}$; Fig. 4e, left), consistent with a motor preparation process where

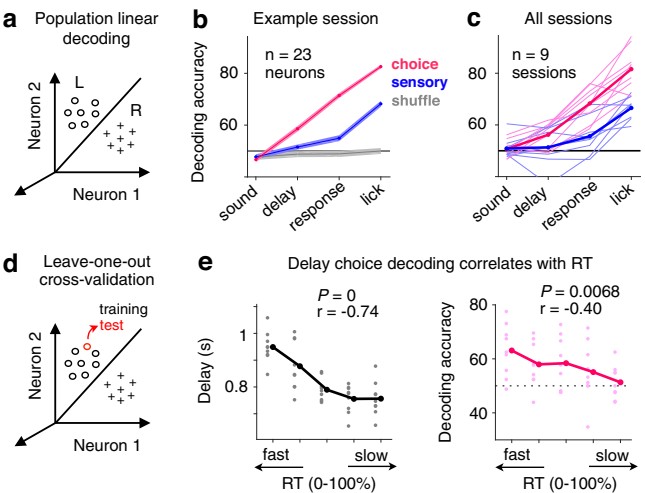

**Fig. 4 Linear classification of population activity predicts animal's choice and RT. a** Schematic to illustrate the linear decoding method using population activity. Clouds of hypothetical population responses corresponding to different conditions (left or right choices) are linearly discriminated in high dimensional neural activity space. **b** Classification accuracy of decoders trained on 23 simultaneously recorded neurons from one example session. Choice decoding corresponds to classification between left versus right choices; sensory decoding corresponds to classification between high versus low click rate trials; gray lines show performance on data with shuffled trial labels. Mean ± s.e.m. across 100 cross-validated samples. **c** Decoding accuracy for all sessions ($n = 9$), similar to **b**. Thin lines, individual sessions; thick lines, mean across sessions. **d** Schematic to illustrate leave-one-out cross-validation. **e** Left, delay durations binned over each 20 percentile of response times (RT) across sessions ($n = 9$). Gray dots, individual sessions; black dots, mean across sessions. Right, delay activity choice decoding accuracy binned over each 20 percentile of RT. Light pink dots, individual sessions; Dark pink dots, mean across sessions. r, Pearson's correlation coefficient, P values report the significance of t tests.

movements are faster when more time is given to prepare for them[27]. Remarkably, the amount of choice information decoded from the SC-projecting M2 populations during the delay period was also negatively correlated with RT ($r = -0.40$, $P < 0.01$; Fig. 4e, right), suggesting that the strength of choice signals accumulated in the M2-SC pathway contributes to motor preparation. We fit a linear mixed effect model (LMM) to simultaneously quantify the effects of delay duration and delay neural activity in predicting RT ("Methods") and found a significant effect of choice decoding accuracy ($P < 0.05$, LMM) beyond the predictive power of delay duration ($P < 10^{-3}$). The effect of decoding accuracy in predicting RT remained the same after controlling for behavioral performance, tongue movement, and nose movement on each trial ("Methods"). Unlike choice-related neural activity, incidental tongue or nose movements during the delay period did not correlate with RT on a trial-by-trial basis ($P$'s > 0.05, LMM; "Methods"). Together, single-neuron and population analyses of projection-based two-photon imaging data reveal that M2 sends progressively stronger choice-related information to SC, and such information during the delay period predicts animals' subsequent behavior.

**Causal contribution of the M2-SC pathway to decision maintenance.** To test whether the choice signals in the M2-SC pathway play a causal role in behavior, we selectively inhibited the projections from M2 to SC using chemogenetics (Fig. 5a). We

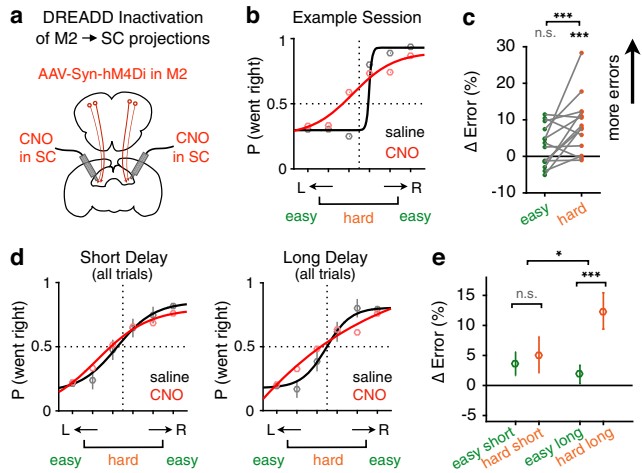

**Fig. 5 Inactivation of M2 terminals in SC impairs decision maintenance. a** Design for bilateral chemogenetic inactivation of M2 terminals in SC. **b** Example SC infusion session (red) compared to saline control session (black) for one animal. Dots, data; lines, sigmoid fit. **c** Difference in error rate between inactivation sessions ($n = 14$) and saline control sessions ($n = 14$) for easy (green) and hard (orange) trials. Gray lines connect data from the same session. ***$P = 4 \times 10^{-4}$ for hard trials; ***$P = 0.001$ for easy versus hard trials; n.s. $P = 0.06$, two-sided permutation test. **d, e** Psychometric curves (**d**) and error rate increase (**e**) due to M2 terminal inhibition in SC, plotted separately for short delay (0.3–0.9 s, $n = 2615$ CNO trials, and 3090 saline control trials) and long-delay trials (0.9–1.5 s, $n = 2357$ CNO trials, and 2704 saline control trials). **d** Mean and s.e.m. across trials. **e** Mean and s.e.m. across 14 pairs of sessions. *$P = 0.0184$; ***$P = 8 \times 10^{-4}$; n.s., $P = 0.61$; two-sided permutation test.

expressed hM4Di in bilateral M2 neurons and locally infused CNO or saline in bilateral SC via implanted cannulae (Supplementary Fig. 8; "Methods"). Different from a general error increase for all trials after M2 soma inactivation (Supplementary Fig. 1d), silencing M2 projections to SC preferentially impaired hard trials but not easy trials ($P < 10^{-3}$; Fig. 5b, c), and led to small changes in other behavioral readouts (Supplementary Fig. 8c, d). This effect is consistent with the difficulty-dependent effect following simultaneous optogenetic inactivation of both M2 and SC neurons during the delay period (Fig. 2p and Supplementary Fig. 4c), suggesting a specific role of the M2-SC pathway in decision-related processes. GLMM fits report a significant decrease in the slope of the psychometric functions ($P < 10^{-4}$), and no significant change in bias ($P > 0.05$). The variable delay period in our task allowed us to further assess the role of the M2-SC pathway on trials when animals needed to maintain decisions over a short versus long period of time. We found that the difficulty-dependent impairment only occurred on long-delay trials (0.9–1.5 s; $P < 10^{-3}$) but not on short delay (0.3–0.9 s) trials ($P = 0.61$; Fig. 5d, e). A repeated-measures two-way ANOVA on error increase revealed a significant main effect of difficulty ($P < 0.05$); a significant interaction between difficulty and delay length ($P < 0.05$); and no effect of delay per se ($P = 0.26$). The delay- and difficulty-dependent impairment supports a critical role of the M2-SC pathway for decision maintenance.

**SC excitatory and inhibitory neurons receive synaptic inputs from M2 and carry choice-related signals**. Our chemogenetic experiment selectively disrupted M2 axons in SC without perturbing other M2 projections. Therefore, the effects we observed here can be specifically attributed to the M2-SC pathway, and

further processing of choice-related information in the SC may be essential for decision maintenance (Supplementary Fig. 4). We thus probed how this information may be transformed within the SC. Combining anterograde transsynaptic viral tracing and immunohistochemistry ("Methods"), we found that 48.3% of the SC neurons downstream of M2 are GABAergic (Fig. 6a), suggesting that M2 terminals target both excitatory and inhibitory neurons in the SC. We confirmed these two types of M2-SC connections by measuring synaptic transmission in genetically labeled excitatory or inhibitory SC neurons using patch-clamp recording while optogenetically stimulating M2 terminals in brain slices (Fig. 6b, c; "Methods"). We observed similar proportions of responsive excitatory (18/24) and inhibitory (28/44) neurons upon M2 axon activation ($\chi^2$ test, $P > 0.05$), with comparable response latency, reliability, and amplitude ($P$'s $> 0.05$, permutation test; Fig. 6d). These data suggest that both excitatory and inhibitory neurons in SC receive synaptic inputs from M2, and may contribute to motor planning in the delayed licking task.

To characterize SC's cell-type-specific functions during task performance, we recorded population-averaged calcium signals from either the excitatory neurons or the inhibitory neurons using fiber photometry in different transgenic mouse lines (Fig. 6e–l; "Methods"). We found that both excitatory and inhibitory SC populations show selective responses beginning from the stimulus period through the delay and execution periods (Fig. 6e–j), and these selective responses mainly encode information for the upcoming action rather than for the sensory cues (Supplementary Fig. 9). To examine how choice-related information may be maintained in different types of SC neurons, we calculated the AUC values that discriminate between correct left and right trials during the stimulus period (early AUC) and the late delay period (late AUC) and examined the variance and stability of choice encoding in these two cell types during motor planning ("Methods"). Overall, we found that SC inhibitory populations were more variable in the strength and preference of choice selectivity compared to SC excitatory populations ($\chi^2$ variance test between cell types, $P < 0.01$ for early AUC; Fig. 6k). Interestingly, although both cell types started to exhibit selectivity during the stimulus period (early AUC), the excitatory populations tended to maintain the same direction of selectivity over time (75.00% sessions with a consistent preference; Fig. 6i, k, l), while the inhibitory populations displayed more dynamic and heterogeneous choice encoding, often showing switched preference from the sound to delay epochs (45.83% consistent sessions; $\chi^2$ test between cell types, $P < 0.05$; Fig. 6j–l). This suggests that the excitatory and inhibitory neurons may play different roles in decision maintenance. In addition, these functional differences between the two cell types parallel the differential connectivity patterns: SC excitatory neurons send descending projections to brainstem nuclei (Gi and IRt) and ascending projections to multiple thalamic nuclei, while also projecting to the contralateral SC, whereas SC inhibitory neurons mainly project to the contralateral SC but avoid the brainstem or thalamus (Supplementary Fig. 10). Together, these cell-type-specific analyses in the SC suggest that the contra-selective motor planning signals are more faithfully maintained over the delay period and transmitted to downstream regions via SC excitatory neurons while the inhibitory SC neurons may play more heterogeneous, modulatory functions within SC.

## Discussion

Our results reveal a critical role of the direct pathway linking the premotor cortex and SC in a memory-guided perceptual decision task. Although earlier studies showed that both the premotor cortex and SC were involved in motor planning[2,4,5,8,18,21], recent

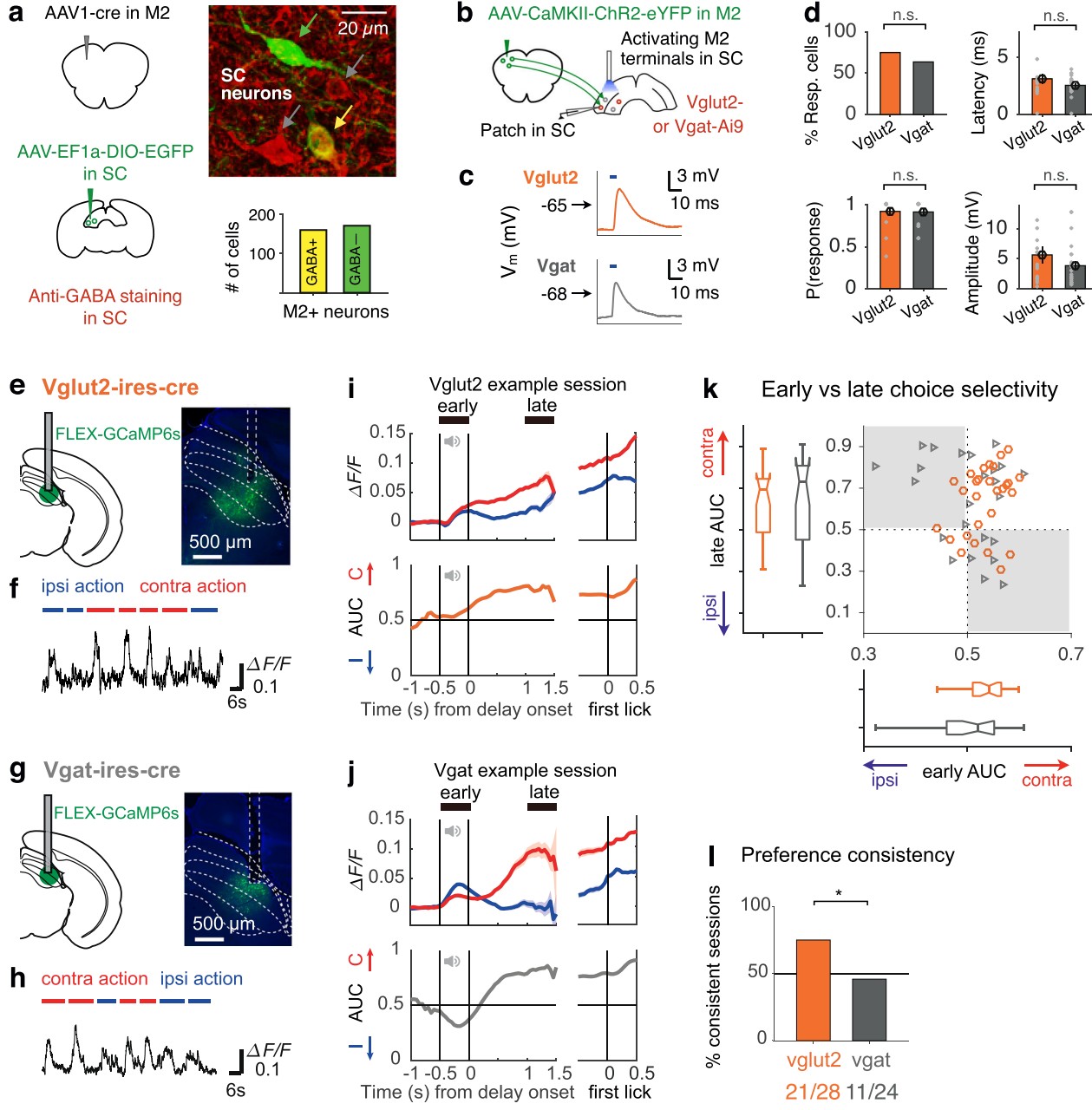

**Fig. 6 SC excitatory and inhibitory neurons receive inputs from M2 and carry choice-related signals. a** Left, SC neurons with M2 inputs are labeled similar to 2**b** (green). GABAergic neurons in SC are labeled using anti-GABA staining (red). Right: green, yellow, and gray arrows point to SC neurons that are M2 + GABA-, M2 + GABA+, or M2− GABA+, respectively. In total, 160 out of 331 M2 + SC neurons are GABA+; data from three mice. **b** Design for cell-type-specific whole-cell recording in SC brain slice during optogenetic activation of M2 terminals. **c** Membrane potential (Vm) changes in example excitatory (Vglut2) and inhibitory (Vgat) SC cells, following a single pulse of photostimulation, indicated by horizontal ticks above the traces. **d** Percent of responsive neurons, response latency, reliability (probability of response), and amplitude for excitatory (Vglut2, $n = 24$) and inhibitory (Vgat, $n = 44$) SC neurons upon M2 terminal activation. Dots, individual cells; error bars, s.e.m. across cells. n.s., not significant; chi-squared or permutation tests, two-sided. **e** Design (left) for fiber photometry experiments to record from SC excitatory populations and example histology that shows the selective expression of GCaMP6s in SC excitatory neurons (right). Six mice were used for this experiment. **f** Example dF/F fluctuations on trials where mice licked ipsi- (blue) or contralaterally (red) to the recording site. **i** Trial-averaged dF/F for ipsi- and contralateral actions (top) and the corresponding AUC value curve (bottom) in one example session; aligned to delay onset (left) or aligned to the first lick during the choice period (right). Centered bin size = 150 ms. AUC values below or above 0.5 indicate a preference for ipsi- or contra-actions, respectively. **g, h, j** similar to **e, f, i**, for SC inhibitory neurons, $n = 5$ mice. **k** Scatterplot: early versus late AUC values for all sessions in SC excitatory populations (Vglut2, $n = 28$ sessions, orange) and SC inhibitory populations (Vgat, $n = 24$ sessions, gray). Boxplot: median, 25th and 75th percentiles of the distributions are indicated by the central mark and edges of each box; lines extend to the furthest data point that is not an outlier. **l** Percent of sessions that maintained the same choice preference during the early and late periods in SC excitatory and inhibitory populations. *$P = 0.031$, chi-squared test, two-sided.

studies have drawn a more complex picture. It was reported that the preparatory activity in the mouse premotor cortex is redundantly and robustly represented in both hemispheres[43], maintained via recurrent excitation in the thalamocortical loop[11], and the resulting motor command was sent to the brainstem[12]. On the other hand, SC is interconnected with all these regions, and gated by the basal ganglia output for initiation of drinking behavior[32]. Simultaneous inactivation of SC and the premotor cortex at specific time points resulted in greater impairment of task performance[5] (Fig. 2), suggesting dynamic cooperation between these two brain regions. However, given the complex networks that both SC and the premotor cortex are embedded in, the underlying circuit logic for decision-related motor planning remains obscure. To solve this, it is crucial to directly examine the projection pathway connecting these two pivotal regions under a well-defined behavioral task with the key variables parametrized and quantifiable. We combined projection-based imaging (Figs. 3 and 4) and inactivation (Fig. 5) methods with a psychometric delayed discrimination task (Fig. 1) to reveal the preferential involvement of the M2-SC pathway in decision-related motor planning. We confirmed that both M2 and SC are critically involved in motor planning, and the cooperation of these two regions during the delay period is crucial. We found that the specific cell type in M2, the SC-projecting neurons, exhibited progressively stronger coding strength for choice information over the trial time. Such dynamic coding was mirrored by the excitatory SC neurons, but not the inhibitory SC neurons, suggesting cell-type-specific interregional coordination. We show that selectively inactivating the M2 projection terminals in SC specifically impaired the decision maintenance during motor planning (Fig. 5). Our experiments thus unravel a mesoscopic circuit mechanism for the premotor-collicular processing underlying decision-related motor planning and represent an effective circuit analysis approach for untangling complex circuit functions.

Although delayed-response tasks with fixed delay duration and simple stimulus–response mappings have been used extensively to discern pathway-specific functions in head-fixed mice[7,12], increased task complexity with parameterized task variables controlled on a trial-by-trial basis becomes necessary to distinguish more nuanced contributions of different circuit elements. Here, we varied both perceptual difficulty and delay duration on a trial-by-trial basis[44]. This task parametrization was crucial in revealing the specific role of the M2-SC pathway in decision maintenance (Fig. 5). Without the cognitively demanding trial types, the causal role of the M2-SC pathway may be erroneously interpreted based on the lack of behavioral deficit on easy trials. Our findings provide a cautionary tale for the interpretation of negative inactivation results.

**Comparison to the primate premotor-collicular circuit.** In the primate oculomotor system, the frontal eye field (FEF) sends direct projections to SC[45,46], and both FEF and SC project to the brainstem saccadic burst generator[47–49], forming parallel premotor-brainstem and premotor-colliculus-brainstem pathways underlying saccade planning and execution. Here, we combined different viral tracing tools to identify a putative SC lick region that receives M2 inputs and projects to the brainstem (Fig. 2a, b and Supplementary Fig. 2), with similar architectural features as the orienting circuit. Functionally, we show that disrupting the information flow in the M2-SC pathway, without perturbing other M2 output projections, selectively impaired decision maintenance (Fig. 5), suggesting a major contribution of the M2-SC-brainstem pathway that cannot be compensated by direct M2-brainstem projections. These data are reminiscent of

previous accounts in primates where the direct pathway from FEF to the brainstem that bypasses the SC only plays a limited role in saccade generation[50]. Our cell-type-specific tracing and recording experiments in the SC further suggest that SC excitatory neurons form the link between the premotor cortex and brainstem during motor planning; whereas the SC inhibitory neurons may play more heterogeneous, modulatory functions (Fig. 6e–l and Supplementary Fig. 10). Future experiments that examine different subtypes of SC inhibitory neurons could provide additional explanations for the functional heterogeneity observed here.

Delay activity sent from the FEF to the SC has been characterized in primates using antidromic stimulation and single-unit recording in memory-guided saccade tasks[51,52]. Similar to our results, these studies reported diverse types of delay signals from the FEF to the SC[51] and found a selective enrichment of movement-related activity in the SC-projecting FEF neurons compared to the general population[52]. Projection-based imaging methods, as we employed here by identifying SC-projecting M2 populations via retrograde viral or beads labeling, facilitate further characterization of this pathway in several ways. First, imaging methods produce a higher yield of projection-specific neurons for statistical analysis (627 M2-SC neurons in our study compared to 66 FEF-SC neurons in ref. [51] and 51 FEF-SC neurons in ref. [52]). Second, antidromic stimulation may excite cortical fibers in the SC other than the premotor-collicular ones and have false-positive cell identification, whereas it is more likely that we only recorded from M2 neurons that directly project to the SC. Third, we were able to sample a large number of premotor-collicular neurons simultaneously, which in turn allowed us to conduct analyses on the population level (Fig. 4). The last point is particularly important because distributed coding mechanisms may be overlooked by only examining individual neurons. For example, Segraves and Goldberg[52] found 4% (2/51) of SC-projecting FEF neurons with anticipatory signals for the upcoming choice, which may suggest that this pathway does not play an important role in motor planning. We found that although most M2-SC neurons were only weakly choice-selective during the delay (Fig. 3g), population-level choice-related information was strong according to linear classification analyses (Fig. 4b, c). Similar results have been observed in a recent study where weakly selective cortical neurons contain strong choice information on the population level[53]. Based on this result, we conducted further analyses using population activity during the delay period (Fig. 4d, e). We found that choice classification accuracy during the delay period can predict animals' subsequent response time on a trial-by-trial basis. On the individual neuron level, Sommer and Wurtz[51] found that the activity of SC-projecting FEF neurons did not predict reaction time, leading to their conclusion that this pathway signaled where to act but not when to act. The difference between our results and these previous findings suggests population-level mechanisms for motor preparation in the M2-SC pathway.

**Movements beyond directional licking.** Although the part of M2 that we investigated is known to be mainly involved in directional licking[7–10], it may be possible that this area contains signals correlated with other incidental movements. Recent studies have shown that ongoing orofacial movements were correlated with brain-wide neural activities both during spontaneous behavior[54] and during a decision-making task[55]. Here, we tracked animals' tongue, forepaw, and nose location during task performance using a video tracking algorithm based on deep neural networks[56]. We found differential movements on the left and right trials beyond directional licking (Supplementary Fig. 7), but these movements were more variable than licking behavior;

contained choice information later than neural selectivity, and did not predict animals' response time on a trial-by-trial basis. Therefore, although it is possible that bodily movements beyond directional licking may be correlated with licking behavior and passively reflect decision variables, these movements could not account for the neural signals we observed. Future studies with an increased resolution for video recording and decoding methods may reveal a more detailed relationship between M2 activity and multidimensional movement.

**Interpretation of inactivation results**. Our unilateral and bilateral perturbations of the M2-SC pathway reveal complementary information for how M2 and SC participate in motor planning (Figs. 2 and 5). Consistent with previous studies that conducted unilateral inactivation in M2 or SC in mice, rats, and monkeys[2,5,8,18,21], our unilateral perturbations support the involvement of these regions in contralateral control of responses, particularly in the movement planning phase (Fig. 2). We did not observe a change in psychometric slopes for unilateral perturbation experiments. This seemingly stimulus-independent effect was also observed in other unilateral inactivation studies (e.g., see refs. [57,58]). It is possible that the strong effects on choice bias after unilateral perturbations may have obscured the measurement of the more subtle changes in perceptual sensitivity or choice memory. Indeed, changes in psychometric function steepness can be more reliably assessed after balanced bilateral inactivations[57,59,60]. It is also possible that unilateral inactivations failed to reveal some behavioral deficits due to compensation from the contralateral hemisphere[43]. Therefore, we also performed bilateral perturbation of M2 terminals in SC and found a significant flattening of the psychometric curves in a memory-dependent fashion (Fig. 5). These results add to the unilateral perturbation data and support the necessary role of the M2-SC circuit during memory-guided licking behavior, even when the left and right hemispheres are balanced.

A growing body of work has observed convoluted relationships between task-relevant activity in a brain structure and the causal role of such activity during behavior[5,61,62]. The size of perturbation effects could positively[7,8] or negatively[5] correlates with the amount of choice information in an underlying area, and the specific timing of temporally precise causal manipulation helps to set critical constraints on mechanistic explanations of decision-making[63,64] and motor planning[10,43]. Here, we found progressively stronger choice information in the M2-SC pathway that peaked during motor execution (Figs. 3 and 4), but the greatest perturbation effects occurred following delay-period inactivation, and not following response period inactivation (Fig. 2). These data suggest that the delay-period activity in the M2-SC pathway is most susceptible to perturbations, revealing the importance of this pathway in maintaining choice memory[65]. Consistent with this interpretation, comparing perturbation effects in task trials with short versus long delays revealed stronger impairment as delay duration increased (Fig. 5). Together, our results present empirical evidence for a distributed premotor-collicular network underlying decision-related processes in motor planning[5], providing a critical foundation for future investigations of cortico-subcortical interaction during cognition and action.

## Methods
**Subjects**. All animal use procedures were approved by the Animal Care and Use Committee of the Institute of Neuroscience, Chinese Academy of Sciences. Fifty-seven adult male mice were used for the experiments presented in this study. Of these, 11 C57BL/6J mice (SLAC) were used for chemogenetic inactivation experiments (8 in the experimental group and 3 in the control group). Nine Vgat-IRES-Cre mice (Jackson Laboratory, JAX016962) were used for optogenetic inactivation experiments (six for behavior testing and three for electrophysiological confirmation of optogenetic effect). Five mice (three C57BL/6J and two Rbp4-Cre,

MMRRC 031125) were used for two-photon calcium imaging experiments. Six mice (two Vglut2-IRES-Ai9, JAX016963 crossed with JAX007909; four Vgat-IRES-Ai9, JAX016962 crossed with JAX007905) were used for whole-cell recordings in brain slices. Eleven mice (six Vglut2-IRES-Cre and five Vgat-IRES-Cre) were used for fiber photometry recording experiments. Fifteen mice were used for anatomical tracing and immunohistochemistry: four Rbp4-Cre mice were used for data in Fig. 2a; two C57BL/6J mice were used for the data in Fig. 2b; three C57BL/6J mice were used for data in Fig. 6a; two C57BL/6J mice were used for data in Supplementary Fig. 2; two Vglut2-IRES-cre mice and two Vgat-IRES-cre mice were used for data in Supplementary Fig. 10.

Mice were group-housed (<6 mice/cage) in a 12 h reverse light:dark cycle, and all experimental procedures were conducted during the dark phase. Mice were water-deprived before the start of behavioral training. On training days, mice received all their water inside training boxes. Supplementary water was provided for mice who could not maintain a stable body weight from task-related water rewards.

**Behavioral apparatus**. In preparation for head-fixed behavioral training, a custom-made head plate was implanted on each mouse during virus infection surgeries. All experiments were conducted in custom-designed and manufactured double-walled, sound-attenuating boxes[66]. Briefly, mice were head-fixed using custom-made holders with their bodies placed in an acrylic tube (25 mm in diameter) and their forepaws gripping the tube edge. Water reward was delivered by two custom-made metal lick ports placed in front of the mouse's mouth. Animal's licks were detected by a capacitive-sensing circuit board connected to the lick ports.

Mice's behavior was controlled by a custom-developed real-time control system (PX-Behavior System)[66], which consists of a custom-designed tone-generating module (TGM) and a microcontroller (Arduino MEGA 2560, IDE 1.5.6r2) for stimulus delivery and measurements of behavioral events. To deliver sound stimuli, TGM sends specified waveforms to an amplifier (ZB1PS, Tucker-Davis Technologies) to drive a speaker (ES1, Tucker-Davis Technologies) placed on the right side of the mice. The sound system was calibrated with a free-field microphone (Type 4939, Brüel and Kjær) to enable a loudness of approximately 70 dB SPL at the position of the mice's ear. Signals from the microphone were digitized with a National Instruments acquisition card (NI 9201) at 500 kHz sample rate for further analysis. Behavioral data were logged using custom-written software in Python (v2.7).

**Behavior**. In the final stage of the behavior (see training procedure below), mice were presented with auditory click trains of different rates (6 stimuli, log spaced between 20 and 125 Hz, 0.5 s). Click rates lower than 50 clicks/s indicated water reward from the left lick port; while higher click rates indicated reward from the right. Stimulus–response mapping was counterbalanced across animals. Each click was a 10 kHz pure tone lasting for 5 ms. Mice needed to discriminate and categorize the sensory cues, form and maintain a motor plan during a variable delay period (randomly sampled from a uniform distribution between 0.3 and 1.5 s), and eventually respond by licking one of the two lick ports after a "go" cue (pure tone, 5 kHz, 40 ms). Trials with premature licking before the "go" cue were terminated immediately, marked as violations, and excluded from further analyses. For non-violation trials, mice were allowed to make their responses within a 3-s window after the "go" cue. Failure to respond within the 3-s window was considered a "miss" trial. Response time (RT) was defined as the time between the onset of the "go" cue and animals' first lick after the "go" cue. Correct responses resulted in ~6 μl of water reward. Error responses were followed by an "error" sound (pure tone, 20 kHz, 1 s), no reward, and a 2–6 s timeout. All trials ended with an intertrial interval (ITI) of 2.5–3.5 s.

The general training procedure was inspired by methods described by Guo et al.[67]. In the first behavioral session, mice received water reward by licking either lick port. In the next 1–2 sessions, the rewarded port alternated between the left and right lick port after three rewards on each side, encouraging the mice to explore both lick ports. Each rewarded lick triggered an auditory stimulus (click trains, 20 or 125 Hz) associated with that side, followed by a 0.3 s delay, the "go" cue, and finally the water reward. The stimulus–response mapping remained constant throughout the training process.

After operant conditioning sessions, mice started two-alternative-forced-choice (2AFC) training and learned to actively discriminate two rates of auditory click trains (20 versus 125 Hz). On each trial, a 0.5-s stimulus was played, followed by a short variable delay (0.3–0.4 s), the "go" cue, and a 3-s response window during which mice's licks were either rewarded or punished. The goal of this training stage was to establish the stimulus–response association; mice were not punished for licking before the "go" cue. We used three methods to facilitate learning. First, during the first 2AFC training session, mice were rewarded for correcting their mistakes within a 1-s window (termed as "grace period") immediately after error responses. Second, error responses were followed by a 2–8 s timeout, during which continued licks to the wrong side reinitiated the timeout period. Third, following each error trial, the same type of trial was repeated once until mice made a correct response on that side.

Mice that reached the criteria of >85% correct on the basic 2AFC task were then trained to withhold licking before the "go" cue. Any lick before the "go" cue (either during the stimulus or delay period) triggered a warning sound (noise stimuli, 1–5

kHz, 0.1 s) and a timeout, during which each additional lick resulted in the same warning sound and reinitiated the timeout period. These trials were terminated immediately and did not lead to any reward. For mice with high performance and low violation rates, delay duration was gradually increased to reach the final range (0.3–1.5 s). In the last phase of behavioral training, trials with intermediate click rates (log spaced between 20 and 125 Hz) were interleaved to test animals' psychometric performance.

Mice trained on the final stage of the task were used for behavioral characterization and chemogenetic manipulation experiments. For optogenetic inactivation experiments, to ensure that all sub-trial inactivation conditions have the same duration for laser stimulation (0.5 s), mice were tested on a modified version of the behavior where the auditory stimulus period and the delay period both lasted 0.5 s instead of a variable delay period as in the original design. For two-photon calcium imaging experiments, we did not include trials with intermediate click rates due to limited trial numbers in imaging sessions.

**Surgery and virus injection**. During surgery, mice were anesthetized with 1–2% isoflurane; and their body temperature was monitored and maintained at 37 °C using a heat pad. Mice were placed in a stereotax with ear bars. The skull was cleaned of blood and tissue and leveled in the anterior-posterior (AP) direction so that the depth difference between bregma and lambda was within 50 μm. The locations of craniotomies for M2 and SC were marked on the skull using the stereotax. After craniotomies were performed, the viral solution was injected into bilateral M2 (±2.5 mm AP, ± 1.5 mm ML from bregma, 0.5 mm below brain surface) or bilateral SC (−3.5 mm AP, ±1.4 mm ML, ±2.3 mm DV from bregma) over the course of ~20 min/0.2 μl injection. The injection system comprised of a custom-made glass pipette (20–30 μm O.D. at the tip; Drummond Scientific, Wiretrol II Capillary Microdispenser) back-filled with mineral oil, and a fitted plunger inserted into the pipette to load or dispense viral solution. The plunger was controlled by a hydraulic manipulator (Narashige, MO-10); and the injection pipette was advanced into the brain using a Sutter MP-225 micromanipulator. After each injection, the glass pipette was left in the brain for 10 min before it was slowly withdrawn, to prevent backflow. Craniotomies were then covered with Kwik-Cast (World Precision Instruments), after which a thin layer of instant adhesive (Loctite 495) was applied onto the remaining skull. Finally, a custom-made head plate was placed on the skull and cemented in place with an appropriate amount of dental acrylic. Post-surgery mice were allowed to recover for at least 3 days before water restriction.

**Chemogenetic inactivation**. We used a DREADD (designer receptor exclusively activated by designer drug) based method[28] to inactivate M2 (Supplementary Fig. 1), or M2 terminals in SC (Fig. 5). AAV2/9-Syn-hM4D(Gi)-mCherry (0.2 μl/site; titer: ~10^13 genomes/mL; Shanghai Taitool Bioscience Co. Ltd) was injected into bilateral M2. For chemogenetic inactivation of M2 terminals in SC, bilateral M2 neurons were infected with the virus and bilateral SC were implanted with cannulae for local infusion. Guide cannula (22 AWG, PlasticsOne, VA) and dummy cannula (same length) were implanted into the brain at a 30-degree angle on each side until the final injector tip (32 AWG, extended 0.5–1 mm beyond the tip of guide cannula) reached −3.5 mm AP, ±1.4 mm ML, and +2.3 mm DV from bregma.

After at least 2 months of virus expression, we conducted intraperitoneal (IP) injection or cannula infusion of Clozapine-N-Oxide (CNO, Sigma) to inactivate infected neurons or terminals before behavioral testing. CNO was dissolved in saline (0.9% NaCl solution) to a stocking solution of 20 mg/mL, and stored as aliquots at −20 °C. For IP experiments, mice were briefly anesthetized with isoflurane, and 100 μl of saline or CNO (1–2 mg/kg) was injected 30 min before each behavioral session. For cannula infusion experiments, mice were either anesthetized with 1–2% isoflurane or head-fixed via the implanted head plate during the infusion process. Dummy cannulae were removed and cleaned with 75% ethanol. Injectors were placed into guide cannulae and extended 0.5–1 mm past the end of the guide. A Hamilton syringe (0.5 μL) connected via tubing to the injector was used to infuse 0.2 μl of saline or CNO (5–10 μg/μL) into each brain area over the course of ~30 s. The injector was left in the brain for 4 min to allow diffusion before removal. For bilateral infusions, the starting side was counterbalanced across sessions. After 4 min, cleaned and rinsed dummies were placed back into the guide cannulae, and mice were removed from isoflurane or head fixation. After 30 min back in their home cages, mice were placed into the behavioral box for testing. Each saline control experiment with stable performance was followed by a CNO experiment in the same animal. The effect of inactivation was quantified as the difference in task performance between the CNO session and the saline control session the day before.

To control for the nonspecific effects of CNO IP injection or infusion, we used a separate group of control mice (n = 3) where M2 neurons were infected with AAV2/9-Syn-mCherry (0.2 μl/site; titer: ~10^13 genomes/mL), and bilateral cannulae were implanted in the SC. All procedures were the same for the experimental and control groups other than the difference in virus infection. We did not observe any significant changes in performance in the control animals (Supplementary Fig. 1i–l and Supplementary Fig. 8e–g), suggesting that the chemogenetic inactivation results we observed in the experimental animals were

not due to the nonspecific effects of CNO injection, but due to the inactivation of neural activity.

**Analysis of behavior and chemogenetic inactivation data**. To characterize animals' baseline psychometric performance (Fig. 1d), we analyzed saline control sessions and non-photostimulation control trials. We combined data across sessions for each mouse and fit that data with a 4-parameter sigmoid function using Matlab's nlinfit. The equation for the 4-parameter sigmoid function is as follows:

$$y = y_0 + \frac{a}{1 + e^{\frac{-(x-x_0)}{b}}} \qquad (1)$$

For these fits, $y$ is "P(lick right)", and $x$ is "log(click rates)" on each trial. The four parameters to be fit are $x_0$, the inflection point of the sigmoid; $b$, the slope of the sigmoid; $y_0$, the lower bound of "P(lick right)"; and $a+y_0$, the upper bound of "P(lick right)". Psychometric curves for individual animals were plotted as thin gray lines (Fig. 1d). We then combined trials across all sessions from all animals and generated a mean psychometric curve (black thick line, Fig. 1d). A similar fitting process was used to generate the psychometric curves for the saline versus CNO sessions in Fig. 5b, d and Supplementary Fig. 1c, k.

To quantify the effect of M2 soma inhibition (Supplementary Fig. 1) and M2 terminal inhibition (Fig. 5) across sessions, we calculated the performance difference between each pair of CNO and saline control sessions. Changes in error rate, violation rate, and miss rate were estimated using the means in each pair of sessions, and changes in RT were estimated using the medians in each pair of sessions. Nonparametric bootstrap procedures or permutation tests were then used to compute significance values across all sessions (shuffled 5000 times).

To investigate how M2 terminal inhibition affected error rates in different trial types, a repeated-measure two-way ANOVA was used to test the within-session main effect of difficulty (easy trials vs hard trials), delay duration (300–900 ms versus 900–1500 ms), and interaction of difficulty and delay on error increase due to inactivation. We also reported paired statistics computed using a permutation test (Fig. 5e). To assess the changes in psychometric functions, we used generalized linear mixed models (GLMM) as implemented in the function "glmer" in package "lme4" (R), in the same way we analyzed optogenetic inactivation results (see analysis of optogenetic inactivation data). We fit a mixed-effects model where mouse's choice on each trial was a logistic function of log(click rates), manipulation condition (CNO or saline control), and their interaction as fixed effects. The mouse and interaction of mouse, log(click rates), and manipulation condition were modeled as within-subject random effects. A significant effect of the manipulation condition represents a change in choice bias; a significant effect of the interaction between log(click rates) and manipulation condition represents a change in the slope of the logistic function. We reported both choice bias and slope changes.

**Optogenetic inactivation**. For optogenetic inactivation of M2 or SC neurons, the viral solution containing AAV2/9-Syn-FLEX-ChrimsonR-tdTomato (0.2 μl/site; titer: ~10^13 genomes/mL; Shanghai Taitool Bioscience Co. Ltd) was injected into bilateral M2 (+2.5 mm AP, ± 1.5 mm ML from bregma, 0.5 mm below brain surface) and bilateral SC (−3.5 mm AP, ±1.4 mm ML, +2.3 mm DV from bregma) of 6 Vgat-IRES-Cre mice. After virus injection, four optic fibers with ceramic ferrule (200 μm in diameter, 0.37 NA, http://www.newdoon.com) were implanted in bilateral M2 (2 mm in length) and bilateral SC (3 mm in length) for a within-subject comparison of inactivated areas. The tips of optical fibers in M2 were pressed against dura, and the tips of optical fibers in SC were positioned 200 μm above the center of SC virus infection site (Supplementary Fig. 3c). Virus expression was allowed to develop for at least 2 months before behavioral testing began.

For each inactivation session, animals' implanted fibers were connected to a red (635 nm) diode-pumped solid-state laser (DPSS; Shanghai Laser & Optics Century Co., Ltd.) via an external optical fiber (200 μm in diameter, 0.37 NA). Laser power at the end of the external fiber was measured with a laser power meter (Sanwa, Mobiken series, LP1) and was adjusted using pulse-width modulation (PWM) to meet the experimental requirements before each session. We used in vivo electrophysiology to search for stimulation parameters for complete inhibition with minimal rebound and minimal residual effect (see next section). For M2, the optimal parameters were 10 Hz laser pulses with 2 ms pulse width at 2 mW laser power measured at the tip of the external fiber (higher frequencies resulted in residual inhibition; Supplementary Fig. 3d, f). For SC, the optimal parameters were 20 Hz laser pulses with 2 ms pulse width at 2 mW laser power (lower frequencies resulted in incomplete inhibition; Supplementary Fig. 3e, g). For simultaneous inactivation of M2 and SC, we used a splitting optical fiber to target both regions at the same time. To ensure the temporal resolution of optogenetic inactivation, we used the lower stimulation parameters (10 Hz, 2 ms pulse width, 2 mW per fiber). Therefore, the simultaneous inactivation corresponded to complete inhibition of M2 and partial inhibition of SC, and can only be directly compared to M2 inhibition alone. Laser illumination occurred on 30% of randomly chosen trials in each behavioral session. Different optogenetic conditions (stimulus, delay, response period, or intertrial interval, 500 ms each) were randomly interleaved for all sessions to control for behavioral fluctuations across days.

The method of using GABAergic activation to silence local activity is well-established in cortex. However, subcortical structures such as the SC contain local

GABAergic interneurons as well as GABAergic projection neurons, some of which project to the contralateral SC[16,68]. Therefore, the network effect of GABAergic activation in the SC may be more complex. Here, we used in vivo electrophysiology to confirm effective and temporally precise silencing of local SC activity (Fig. 2d and Supplementary Fig. 3e, g). In addition, we observed an ipsilateral bias after unilateral SC inactivation, consistent with a more dominant effect of silencing local activity. Future experiments that selectively label GABAergic projection neurons versus local interneurons are needed to test their differential roles during motor planning.

**Electrophysiological verification of optogenetic effects**. To measure the effects of optogenetic inactivation on neural activity, acute recordings of infected M2 or SC neurons were performed in awake mice using custom-made optrodes (Supplementary Fig. 3d–g). A sharp tungsten electrode (0.5 or 1.0 MΩ, World Precision Instruments, Inc.) was glued to a stripped optical fiber (200 μm in diameter, 0.37 NA), with the tip of the optical fiber 200–300 μm above the tip of the tungsten electrode to parallel conditions in optogenetic inactivation experiments. The optrode was then threaded into a glass tubing for protection; glued in position with epoxy so that <1 cm of the optrode protruded from the glass tubing. During acute recording sessions, the optrode was advanced to the center of the infected area in awake head-fixed mice. For each neuron tested, baseline neural activity was recorded for 4 s, followed by 0.5 or 3 s of laser stimulation, and another period of post-stimulation recording, repeated 20 times. We systematically tested different laser power, laser pulse width, and pulse frequency to find optimal stimulation parameters for temporally precise inactivation of M2 and SC neurons.

**Analysis of optogenetic inactivation data**. To estimate the effect size of the ipsilateral bias due to optogenetic inactivation, we compared P(lick ipsi) on inactivation trials and control trials from the same sessions. For each session, we calculated the mean P(lick ipsi) on control trials and subtracted that mean value from the P(lick ipsi) of individual inactivation trials. After obtaining the normalized ipsilateral bias change for trials in each session, we concatenated trials across all sessions and all mice, and computed the mean and s.e.m. of delta ipsilateral bias across trials.

To visualize the effects of optogenetic inactivation on animals' psychometric performance, we fit a 4-parameter sigmoid function to control and inactivation trials across all sessions for each inactivation epoch (Fig. 2g, j, m), similar to the fitting process used to generate Fig. 1d. These fits are for visualization only. All statistics reported for optogenetic inactivation experiments were computed using generalized linear mixed models (GLMM) as implemented in the function "glmer" in package "lme4" (R). For unilateral M2 (Fig. 2g) or SC (Fig. 2j) inactivations, we fit a mixed-effects model where mice's choice on each trial was a logistic function of log(click rates), optogenetic inactivation epoch (ITI, stimulus, delay, response period, or no inactivation), and their interaction as fixed effects. The mouse and interaction of mouse, log(click rates), and optogenetic inactivation time were modeled as within-subject random effects. The statistics reported for M2 or SC inactivations were the fixed effect P values for the ipsilateral bias change due to optogenetic inactivation in that specific epoch compared to the no inactivation condition. We then conducted multiple comparisons ("mcp" in "multcomp", R) of the GLMM results to simultaneously test whether the delay period inactivation effect was significantly different from the stimulus period effect and the response period effect. Adjusted P values from this single-step method were reported. The significance of the interaction between log(click rates) and optogenetic inactivation, which represents the change in slope of the logistic function, was also reported.

To compare inactivation results in different brain regions at different behavioral epochs, we used a GLMM to fit combined inactivation trials from M2 inactivation, SC inactivation, and simultaneous M2 and SC inactivation during the ITI, stimulus, delay, and response period (Model 1, Supplementary Fig. 4). In this model, mice's choice on each trial was a logistic function of log(click rates), optogenetic inactivation epoch, and optogenetic inactivation region as fixed effects and within-subject random effects. We reported adjusted P values of the multiple comparison results that tested if the stimulus, delay, response period effects were significantly different from each other; and if the M2, SC, or M2+SC inactivation effects were significantly different from each other.

To test how optogenetic inactivation affected hard versus easy trials differently, we included a three-way interaction term (epoch: region: difficulty) to Model 1 described above (Model 2, Supplementary Fig. 4). The difficulty was defined as a quadratic function of the normalized log(click rates). Defining difficulty as 1 minus the absolute value of the normalized log(click rates) did not change the main effects. We demonstrated that Model 2 was a significantly better fit to the data than Model 1 using a likelihood ratio test ("ANOVA", R). Fixed effect P values for the interaction terms were reported.

**Two-photon calcium imaging**. Two methods were used to selectively label M2 neurons that project to the SC. For three C57BL/6J mice, AAV2-Retro-Cre (titer: ~10^13 genomes/mL; diluted threefold in saline; 0.2 μL; Shanghai Taitool Bioscience Co. Ltd) was injected into the left SC (−3.5 mm AP, −1.4 mm ML, +2.3 mm DV from bregma). After behavioral training, a circular craniotomy (~2.5 mm in diameter) was made above the left M2 (centered at +2.5 mm AP, −1.5 mm ML from

bregma), and AAV2/9-Syn-FLEX-GCaMP6s (titer: ~10^13 genomes/mL; diluted threefold in saline; Shanghai Taitool Bioscience Co. Ltd) was injected at 3–5 locations (50 nl per site) at a depth of 0.5–0.6 mm from the brain surface. With this method, only SC-projecting M2 neurons would express GCaMP virus; and both apical dendritic trunks (Supplementary Fig. 6a–c) and somas (Supplementary Fig. 6d–f) were imaged at different sessions. For one Rbp4-Cre mouse, red RetroBeads (diluted fivefold in saline; 0.2 μL; Lumafluor) were injected into the left SC, and AAV2/9-Syn-FLEX-GCaMP6s (titer: ~10^13 genomes/mL; diluted threefold in saline; Shanghai Taitool Bioscience Co. Ltd) was injected into the left M2. With this method, only GCaMP-expressing M2 somas with beads co-localization were identified as SC-projecting M2 neurons. All GCaMP-expressing M2 somas from two Rbp4-Cre mice (layer 5 M2 neurons without projection specificity; Supplementary Fig. 6g–i) were used to compare to SC-projecting M2 neurons. Custom-designed imaging window (Fig. 3a) was constructed from an outer steel ring, a cannula tube (0.6 mm height), and one layer of microscope coverglass (2.3 mm in diameter) glued to the outer surface of the cannula tube. During the first surgery, a virus or RetroBead was injected into the SC, and the location of M2 craniotomy was marked on the skull before a custom-made head plate was implanted. After behavioral training, mice underwent the second surgery, where M2 craniotomy and GCaMP viral injection were performed before fixing the imaging window to the skull using dental acrylic. Water restriction started 3–5 days after each surgery.

Imaging experiments started 3–4 weeks after GCaMP viral infection. Images were acquired using a custom-built two-photon microscope (http://openwiki.janelia.org/wiki/display/shareddesigns/MIMMS), with a resonant galvanometer (Thorlabs; 16 kHz line rate, bidirectional). GCaMP6s was excited using a Ti-Sapphire laser (Coherent) at 920 nm, and the average laser power for imaging SC-projecting M2 somas and apical trunks was ~160 and ~120 mW, respectively. Emitted fluorescence was isolated using a bandpass filter (525/50, Semrock). The objective was a ×16 water immersion lens (Nikon, 0.8 NA, 3 mm working distance). The field of view was 300 by 300 μm (512 × 512 pixels), imaged at ~30 frames/s. The entire microscope was enclosed in a custom-designed sound attenuation box, and the system was controlled using ScanImage (http://scanimage.org). Images were acquired continuously for the entire session. To identify the M2 neurons retrogradely labeled with Retrobead, a small image stack was acquired around the imaging depth at the end of each functional imaging session (laser wavelength = 840 nm).

**Video tracking and movement estimation during two-photon calcium imaging**. Infrared LEDs were used to enable video acquisition in darkness during imaging sessions. Videos were recorded using a camera (KS2A418, kingcent) at 24 Hz. We estimated animals' motion using a video tracking algorithm based on deep neural networks (DeepLabCut)[56]. The model was trained to estimate the horizontal and vertical locations of animals' tongue, left and right forepaws, nose, and left and right lick ports during the imaging sessions. For further analyses, we only included video frames with trained model likelihood >0.1 (Supplementary Fig. 7b).

For each session, we calculated the baseline position for each tracked item (for example, the horizontal position of the tongue) by averaging the coordinates for that item over the entire session. This baseline was then subtracted from all the frames to get a change in position for that item (Δpixel) over time. To test when the changes in position (movement) differentiated between trial types, we performed a ROC analysis on video data to distinguish trials where animals licked ipsi- (positive Δpixel values) or contralaterally (negative Δpixel values) to the imaged side as a function of time (bin = 2 video frames, ~80 ms). The significance of AUC was determined by shuffling trial labels 1000 times and testing if the AUC value was out of the 99% confidence interval for the shuffled data. To compare the timing of earliest choice selectivity based on neural population responses (SVM) and based on tongue movement, we calculated the difference in tongue movements between ipsi- and contra-action trials as a function of time for each session and computed the first time bin where such difference was significantly different from 0 across sessions (P < 0.01, bootstrap). Delay-period neural activity encoded upcoming choices hundreds of milliseconds earlier than any video-detected tongue movements (~400 ms versus ~1000 ms after delay onset; Supplementary Fig. 7c, d).

**Analysis of two-photon calcium imaging data**. Brain motion was corrected using a cross-correlation-based registration algorithm[69]. Regions of interest (ROIs) were manually selected based on identifiable cell bodies and dendritic apical trunks. The boundaries of the ROIs were determined using mean and maximum fluorescence intensity across frames of all trials. The fluorescence (F) time series of each ROI was estimated by averaging all the pixels within the ROI. Slow timescale fluorescence changes were corrected by determining the distribution of fluorescence values over 400 frames (~14 s) around each time point and subtracting the eighth percentile value[70]. ΔF/F₀ was then calculated for each ROI as (F−F₀)/F₀, where F₀ is the mode of F over the entire session after slow timescale fluorescence correction. For each trial, averaged fluorescence signal before sound onset was subtracted from activity at each frame.

Due to the variable delay and response durations across trials, we could not rely on any one-time point alignment to fairly compare neural selectivity across all behavioral epochs. Instead, we first extracted and averaged neural responses during the relevant time window for each behavioral epoch per trial (Fig. 3e). For variable delay durations, activity within 500 ms before the "go" cue was taken for each trial.

Response period activity was extracted from the entire response time, which varied across trials. Using these epoch-averaged fluorescence values (Fig. 3f), single-neuron and population-level analyses were conducted for all behavioral epochs across trials.

To quantify single-neuron selectivity, we conducted receiver operating characteristic (ROC)[42] analysis on responses in each behavioral epoch and calculated the area under the ROC curve (AUC) to indicate discrimination between trial types. Correct and error trials were included to calculate choice selectivity (lick left vs lick right trials) and sensory selectivity (high-rate vs low-rate trials) for each neuron. To compute the significance of AUC values, we shuffled the trial labels 1000 times to obtain a distribution of AUC values for shuffled data. For a given neuron, if the sensory or choice AUC was outside the 99% confidence interval of the shuffled data during any behavioral epoch, that cell was labeled as significantly side-selective (Fig. 3b, c).

To characterize the changes in choice AUC values across behavioral epochs (Fig. 3g and Supplementary Fig. 5b, c), we converted each neuron's choice AUC value to a range between 0 and 1, where 0 represents a strong preference for ipsilateral choices, 1 represents a strong preference for contralateral choices, and 0.5 represents no choice preference. For each behavioral epoch, the histogram and gaussian fit to the distribution of converted choice AUC values for all individual neurons were plotted for one example animal (Fig. 3g) or for all animals (Supplementary Fig. 5b, c). Individually significant AUC values ($P < 0.05$) were marked in black. For each pair of epochs (sound versus delay, delay versus response, response versus lick), a nonparametric permutation test (shuffled 5000 times) was used to compute statistical difference between distributions of choice AUC values ($P < 0.01$ for all pairs in the example animal, $P < 10^{-3}$ for all pairs for data across all animals).

To measure the amount of choice or sensory-related information in simultaneously imaged neural populations (Fig. 4a–c), we performed a series of cross-validated linear classification analyses (support vector machine, SVM, "fitcsvm", Matlab). For each session, simultaneously imaged calcium signals were arranged in a $M \times N \times T$ matrix, where $M$ is the number of trials, $N$ is the number of neurons, and $T$ is the number of time points (task epochs). For a given trial and one task epoch, the vector of $N$ neurons' $\Delta F/F_0$ values represents the population response vector for that condition (Fig. 4a). To decode choice or sensory-related information, we randomly sampled 90% of imaged trials as the training set, and the remaining 10% of trials as the test set. The training set was used to compute the linear hyperplane that optimally separated the population response vectors corresponding to lick-left versus lick-right trials (choice decoding) or high-rate versus low-rate trials (sensory decoding). The number of training trials were balanced between conditions to avoid bias in favor of one condition over the other. Performance was calculated as the fraction of correct classifications of the test trials. Decoders were trained and tested independently for each behavioral epoch. We repeated the resampling process 100 times and computed the mean and standard error of decoding accuracy across the 100 resampling iterations. To test if one session's decoding accuracy was significantly higher than chance (0.5), or if choice decoding was significantly better than sensory decoding, we used bootstrap or permutation tests.

To investigate the relationship between animals' response time (RT) and the amount of choice information during the delay on a trial-by-trial basis, we used leave-one-out cross-validation (Fig. 4d, e). For each session with $M$ trials, the training and testing procedure was repeated $M$ times. For each iteration, we trained the decoder on all but one trial (balanced between conditions), and tested the classification performance (0 or 1) for that one test trial. We then binned trials based on RT (0–2.5 s), and calculated the mean choice decoding accuracy for each 20 percentile of RT per session (Fig. 4e, right). The relationship between decoding accuracy and RT was quantified using linear correlation. Because delay duration and delay-period neural activity were both negatively correlated with RT, we fit a linear mixed model (LMM, "fitlme", Matlab) to simultaneously consider their contributions for predicting RT on a trial-by-trial basis. Animals' RT on each trial was modeled as a linear combination of delay duration and the classification performance for that trial, both as fixed effects and as within-subject random effects. The LMM statistics reported were the fixed effect $P$ values for predicting RT (delay duration or choice decoding accuracy). Including behavioral performance on each trial (correct or error) as a third fixed effect in the LMM did not change the main conclusions. Finally, we fit a LMM where RT was modeled as a linear combination of delay duration, delay period neural classification performance, behavioral performance, tongue location (during the last 500 ms of the delay), and nose location on each trial, both as fixed effects and as within-subject/side random effects. We did not find any significant effect of tongue or nose movement for predicting RT.

**Slice preparation and recording.** To study how M2 activity modulates different types of SC neurons, we performed ex vivo slice electrophysiology recording. We injected AAV-DJ-CaMKIIa-ChR2-EYFP virus into unilateral M2 (+2.5 mm AP, ± 1.5 mm ML from bregma, 0.5 mm below brain surface, 0.5 μL/site; titer 1.81E14 v. g./ml; BrainVTA Co. Ltd) and recorded SC neurons' responses in acute brain slices upon optogenetic stimulation of M2 terminals. Transgenic mice were used to label excitatory (Vglut2-IRES-Ai9, JAX016963 crossed with JAX007909) and inhibitory (Vgat-IRES-Ai9, JAX016962 crossed with JAX007905) neurons. Three to 4 weeks after virus injection, mice (~10 weeks of age) were sacrificed and coronal brain

slices (300-μm thickness) were acquired with vibratome (LEICA VT 1200 s) in ice-cold recovery solution (containing 93 mM NMDG, 2.5 mM KCl, 1.2 mM $NaH_2PO_4$, 30 mM $NaHCO_3$, 20 mM HEPES, 25 mM glucose, 5 mM sodium ascorbate, 2 mM thiourea, 3 mM sodium pyruvate, 10 mM $MgSO_4$, 0.5 mM $CaCl_2$). Sections were immediately transferred to recovery solution at 32°C for 5 min and then incubated in ACSF (containing 125 mM NaCl, 2.5 mM KCl, 1.25 mM $NaH_2PO_4$, 1 mM $MgCl_2$, 2 mM $CaCl_2$, 12.5 mM glucose, 25 mM $NaHCO_3$) at room temperature till recording.

To find regions with a high density of M2 terminals and target tdTomato+ cells, we examined the expression pattern in each brain slice under a fluorescence microscope (Olympus U-TV1 X). Borosilicate glass pipettes containing intracellular solution (135 mM K-gluconate, 5 mM KCl, 10 mM HEPES, 0.3 mM EGTA, 4 mM MgATP, 0.3 mM $Na_2$GTP, 10 mM $Na_2$-phosphocreatine) with impedance larger than 3 MΩ were used to patch neurons. Signals were recorded with a MultiClamp 700B amplifier (Axon Instrument), under the current-clamp mode, filtered at 2 kHz, and sampled at 10 kHz. Single light pulse (5 ms, 25 mW power from the objective) or a train of ten pulses (10 Hz, each pulse 5 ms) were delivered to activate M2 terminals using a mercury arc lamp via the objective of the microscope.

**Analysis of slice recording data.** To determine if an identified excitatory or inhibitory SC neuron is responsive to M2 terminal activation, we repeatedly delivered trains of 10-pulse photostimulation and calculated the rate of significant responses based on average EPSPs over repeated trials. Light evoked voltage changes (within 50 ms after light delivery) exceeding 3 s.d. of baseline voltage (over 1 s before stimulation) were considered significant transients, and cells with >40% significant transients were considered responsive. A chi-squared test was used to calculate the statistical difference between the percentage of responsive inhibitory neurons versus excitatory neurons.

For responsive neurons, we repeatedly delivered single-pulse photostimulation to further characterize their response reliability, latency, and amplitude. Reliability was calculated as the probability of significantly responsive trials for each neuron. Response latency and amplitude were calculated using mean response traces averaged over all individual trials. The latency of EPSP corresponded to the amount of time between light stimulation onset and the first time point when voltage change exceeded 3 s.d. of baseline. Response amplitude corresponded to the maximum voltage change above baseline within 50 ms after light delivery. Permutation tests (shuffled 5000 times) were used to report the statistical differences between two groups of SC neurons.

**Fiber photometry recordings.** To compare activities of SC excitatory neurons and inhibitory neurons during task performance, we expressed AAV2/9-hSyn-FLEX-GCaMP6s (0.2 μL/site; titer: ~$10^{13}$ genomes/mL; Shanghai Taitool Bioscience Co. Ltd) in bilateral SC (−3.5 mm AP, ± 1.4 mm ML, +2.3 mm DV from bregma) of either Vglut2-IRES-Cre mice ($n = 6$) or Vgat-IRES-Cre mice ($n = 5$). After virus injection, we implanted optic fibers with ceramic ferrule (200 μm in diameter, 3 mm in length, 0.37 NA, http://www.newdoon.com) above the SC (2 mm DV from dura). Virus expression was allowed to develop for at least 2 months before recording began.

For each recording session, blue light from a 470-nm LED (XLamp XQ-E Blue Starboard, Cree Inc.) was bandpass filtered (FB470-10, Thorlabs) and delivered into the SC through an optic fiber (NA 0.37, 200 μm core) to excite GCaMP6s (~10 μW at the fiber tip). To control for fluorescence change caused by movement, another light source from a 410-nm LED (Xuming, 410–420 nm) was bandpass filtered (FB410-10, Thorlabs) and delivered to the same site to measure auto-fluorescence. The power of 410-nm excitation light was adjusted so that the baseline 410-nm emission light intensity matched that of the 470-nm emission light. Alternating 470-nm and 410-nm fluorescent signals were collected by the same light pathway and recorded using a CCD camera (Retiga R1, Teledyne QImaging), with an effective frequency of 20 Hz/channel. A custom-written LabVIEW (https://www.ni.com) GUI was used to control excitation light delivery and camera recording through a multifunction I/O device (National Instruments, USB-6341). Micro-Manager (Version 2.0 beta) was used to save emission images from the camera as TIFF files. The initiation of each trial was synchronized between imaging acquisition and behavior by a trigger signal sent from our in-house behavioral system to the I/O device. Whenever possible, fluorescence changes in bilateral SC were recorded simultaneously.

**Analysis of fiber photometry data.** Slow timescale fluorescence fluctuations for GCaMP6s signal (470-nm) and auto-fluorescence signal (410-nm) were respectively corrected by determining the distribution of fluorescence values over 1000 frames (~50 s) around each time point and subtracting the fifth percentile value. We conducted motion correction by first transforming the slow-corrected auto-fluorescence signal to fit the slow-corrected GCaMP6s signal via linear regression, and then subtracting this fitted control signal from the GCaMP6s signal. $\Delta F/F_0$ was calculated as $(F–F_0)/F_0$, where $F_0$ is the mode of fluorescence $F$ over the entire session after slow correction and motion correction. For each trial, averaged fluorescence signal before sound onset was subtracted from activity at each frame.

To quantify neural selectivity, we calculated the area under the ROC curve (AUC) on responses to indicate discrimination between trials where mice licked

ipsi- or contralaterally from the recording site (correct trials only). For the example session traces (Fig. 6i, j), activity on different trials was smoothed (overlapping bins of three frames) and aligned to delay the onset or to first lick; AUC values were then calculated across trials as a function of time. Activity after delay offset for short delay trials was masked for the delay-period analysis. We converted the AUC values to a range between 0 and 1, where 0 represents a strong preference for ipsilateral choices, 1 represents a strong preference for contralateral choices, and 0.5 represents no choice preference.

For across-session analyses in Fig. 6k, l, we extracted and averaged neural activity during the sound period (early epoch) and 1–1.5 s after delay onset (late epoch) for each trial and calculated AUC values across trials during the early and the late epochs for each session. The direction (prefer ipsi or contra) and strength (strong or weak preference) of selectivity across sessions were then used to compare the coding patterns in SC excitatory versus inhibitory populations. Error trials were included for analyses that examined sensory versus choice selectivity during the late epoch (Supplementary Fig. 9).

**Tracing experiments**. All viruses used in tracing and immunostaining experiments were provided by Shanghai Taitool Bioscience Co. Ltd and had a titer of ~$10^{13}$ genomes/mL. To identify the part of SC downstream of M2 (Fig. 2a), we unilaterally injected AAV-Syn-FLEX-EGFP (0.2 μL/site) into M2 (+2.5 mm AP, ± 1.5 mm ML from bregma, 0.5 mm below brain surface) of Rbp4-Cre mice, which constrained virus expression to M2 layer 5 neurons.

To label SC somas that receive M2 projections, we applied anterograde transsynaptic tracing (Fig. 2b) using one serotype of adeno-associated virus (AAV1)[33]. AAV1-hSyn-Cre (0.2 μL/site) was unilaterally injected into M2 of C57 mice, and AAV-hSyn-FLEX-tdTomato (0.2 μL/site) was injected into the ipsilateral SC (−3.5 mm AP, ± 1.4 mm ML from bregma, 2.3 mm below bregma).

To compare the projections of M2 and SC in the brainstem, we simultaneously labeled M2 neurons and SC neurons downstream of M2 (Supplementary Fig. 2) with different fluorescence. We injected a mixture of virus-containing AAV1-hSyn-Cre and AAV-hSyn-FLEX-tdTomato (volume ratio, 1:1, 0.1 μL/site in total) unilaterally into the M2 of C57 mice. In the ipsilateral SC, we also injected AAV-CAG-FLEX-EGFP (0.2 μL/site). Red and green terminals in the brainstem represent direct projections from M2 and projections from SC neurons downstream of M2, respectively.

To compare the projections of SC excitatory versus inhibitory neurons (Supplementary Fig. 10), we separately injected pAAV-efla-DIO-mCherry (0.2 μL/site) into Vgat-IRES-cre or Vglut2-IRES-cre mice ($n = 2$ animals for each experiment).

For all experiments mentioned above, mice were sacrificed 4 weeks after virus injection. Brain slices were imaged using an Olympus VS120 Virtual Slide Fluorescence Microscope.

**Immunostaining**. To characterize the types of SC neurons that receive M2 projections, we combined anterograde transsynaptic viral tracing with immunohistochemistry (Fig. 6a). We injected AAV1-hSyn-Cre (50nL/site) unilaterally into M2 and AAV-EF1a-DIO-EGFP (0.2 μL/site) into the ipsilateral SC. SC neurons downstream of M2 were thus labeled with green fluorescence. Four weeks after virus injection, mice were perfused using 4% paraformaldehyde (PFA). Brains were incubated in PBS solution containing 30% sucrose overnight at room temperature and coronally sectioned with a cryostat (20 μm). Slices containing SC neurons were collected and underwent immunostaining.

After incubated with blocking buffer (PBS containing 5% bovine serum albumin and 0.3% Triton X-00) for 2 h at 37 °C, sections were incubated with the primary anti-GABA antibody (rabbit, dilution 1:1000, A2052; Sigma) overnight at 4 °C. The primary antibody was then washed three times with PBS (15 min each) before adding the secondary antibody (donkey anti-rabbit, Alexa Fluor 594, dilution 1:1000, A-21207; ThermoFisher Scientific). Brain sections were incubated in the secondary antibody for 2 h at 37 °C. Finally, the secondary antibody was washed three times (15 min each) with PBS, and sections were mounted onto microscope slides with a reagent containing DAPI to stain nuclei. Sections were imaged with Nikon A1 inverted confocal microscope (×20 objectives) and confocal images were analyzed with ImageJ.

To quantify the percentage of GABA+ cells (red) in total SC neurons that receive M2 projections (green), we sampled three slices (−3.28 to −4.04 mm AP from bregma) for each animal and manually counted EGFP-labeled SC neurons. Among these SC neurons downstream of M2, we checked whether they were GABA immunoreactivity positive (red) to calculate the ratio of GABAergic versus non-GABAergic neurons.

**Histology**. Histology was performed on all mice used for imaging, electrophysiology, chemogenetic, and optogenetic experiments after data collection was completed. Animals were anesthetized and perfused with 4% PFA or with 10% formalin, and brains were removed and post-fixed overnight. Brains were then incubated with PBS solution containing 30% sucrose overnight. Frozen sections (50 μm) containing viral injection sites, cannula implantation tracks, or optic fiber tracks were gathered and stained with DAPI to visualize nuclei. Pictures were taken with an Olympus MacroView MVX10 fluorescent microscope.

For the chemogenetic inhibition experiments, six out of eight mice that underwent surgeries had cannulae placed correctly within the SC. These six mice then included for analyses in the M2 terminal inhibition experiment (Supplementary Fig. 8b). The two mice that had cannulae placed outside of the SC were only included for analyzing the M2 soma inhibition experiment but excluded for the terminal inhibition experiment. For the optogenetic inhibition experiments, all six mice were included as the final dataset based on effective virus expression and correct optic fiber placements (Supplementary Fig. 3c).

**Reporting summary**. Further information on research design is available in the Nature Research Reporting Summary linked to this article.

## Data availability
All data needed to understand and assess the conclusions of this study are available in the main text or the Supplementary Materials. All the original behavioral, optogenetic, chemogenetic, electrophysiological, imaging, and tracing data are archived in the Institute of Neuroscience, Chinese Academy of Sciences, and can be obtained upon reasonable request via email to the corresponding authors.

## Code availability
All data acquisition and analysis codes are archived in the Institute of Neuroscience, Chinese Academy of Sciences, and can be obtained upon reasonable request via email to the corresponding authors.

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

## Acknowledgements

We thank J. Pan for technical support; E. Huang for help with the immunostaining experiments; J.C. Erlich for advice regarding the GLMM; A.T. Piet and C.D. Kopec for discussion regarding the dynamical attractor model literature; N. Li and K. Svoboda for discussion regarding the comparison between head-fixed mouse and freely-moving rat studies; X.H. Xu, X.K. Chen, and Z.C. Guo for comments on the manuscript; and other members of the Xu lab for help and advice during all stages of the project. This work was supported by the National Key R&D Program of China, Grant No. 2017YFA0103900/ 2017YFA0103901; the CAS-NWO International Cooperation Project of the Chinese Academy of Sciences, Grant No. 153D31KYSB20160081; the "Strategic Priority Research Program" of the Chinese Academy of Sciences, Grant No. XDB32010000; National Natural Science Foundation of China, Grant No. 31571081; NSFC-ISF International Collaboration Research Project, Grant No. 31861143034; Key Research Program of Frontier Sciences, CAS, Grant No. QYZDB-SSW-SMC045; Shanghai Municipal Science and Technology Major Project, Grant No. 2018SHZDZX05; the Youth Thousand Talents Plan (to N.L.X.). C.A.D. is supported by the Simons Collaboration on the Global Brain Postdoctoral Fellowship and the CPSF-CAS Joint Foundation for Excellent Postdoctoral Fellows.

## Author contributions

C.A.D. and N.L.X. conceived the project and designed the experiments. C.A.D. developed the behavior and performed the optogenetic and imaging experiments. Y.P. performed the chemogenetic, fiber photometry, and immunostaining experiments. Y.P. and T.Z. performed the tracing experiments. Y.P., G.M., and S.Z. performed the slice electrophysiology experiments. C.A.D. and Y.P. performed the data analyses. C.A.D., N.L.X., and Y.P. wrote the manuscript.

## Competing interests

The authors declare no competing interests.
