## [Peer Review File · Nature Communications]

Reviewers' Comments:

Reviewer #1:

Remarks to the Author:

I thank the authors for their detailed revision and response to my comments. I am satisfied with their responses and do not have additional comments. I congratulate them on this study.

Reviewer #3:

Remarks to the Author:

The authors thoroughly responded to all my concerns. The revised manuscript is much improved and now ready to be published in Nature Communications.

Response to Referees on “A cortico-collicular pathway for motor planning in a memory-dependent perceptual decision task”

Black, reviewer comments

Blue, our response

Reviewer #1:

I thank the authors for their detailed revision and response to my comments. I am satisfied with their responses and do not have additional comments. I congratulate them on this study.

We thank the reviewer for this positive evaluation of our work, and for all the reviewer's comments that helped us improve our manuscript.

Reviewer #3:

The authors thoroughly responded to all my concerns. The revised manuscript is much improved and now ready to be published in Nature Communications.

We thank the reviewer for this positive evaluation of our work, and for all the reviewer's comments that helped us improve our manuscript.